# Quality Analysis of Tuberculosis Specimens Transported by Drones versus Ground Transportation

**Diosdélio Malamule [1,\*], Susana Moreira [2], Carla Madeira [1], Carla Lutucuta [2], Gabriella Ailstock [3]** **, Luciana Maxim [3], Ruth Bechtel [2], Olivier Defawe [3] and Sofia Viegas [1]**

[1]   Instituto Nacional de Saúde, Vila de Marracuene, EN1, Marracuene 3943, Mozambique; carla.madeira@ins.gov.mz (C.M.); sofia.viegas@ins.gov.mz (S.V.)
[2]   VillageReach, Rua 1.426, Avenida para o Palmar nº 84, Maputo 1100, Mozambique; susana.berjano.moreira@gmail.com (S.M.); carla.lutucuta@villagereach.org (C.L.); ruth.bechtel@villagereach.org (R.B.)
[3]   VillageReach, 210 S Hudson St., Suite 307, Seattle, WA 98134, USA; gabriella.ailstock@villagereach.org (G.A.); luciana.maxim@villagereach.org (L.M.); olivier.defawe@villagereach.org (O.D.)
\*   Correspondence: diosdelio.gwegwe@ins.gov.mz; Tel.: +258-842-196-330

**Abstract:** There are many challenges that impact the current referral network for Tuberculosis (TB) sputum specimens in Mozambique. In some cases, health facilities are remote and the road infrastructure is poor and at times impassable, leading to delays in laboratory specimen transportation and long turn-around times for results. Drone transportation is a promising solution to reduce transportation time and improve access to laboratory diagnostics if the sample quality is not compromised during transport. This study evaluated the impact of drone transportation on the quality of TB sputum specimens with suspected Mycobacterium tuberculosis. 156 specimens were collected at five (5) health centers and sent to the Instituto Nacional de Saúde (INS) National TB Reference Laboratory. Specimens were then equally divided into two aliquots; one to be transported on land and the other by air using a drone. Control and study group specimens were processed using the NALC-NaOH method. Agreement between sample and control specimens was acceptable, indicating that drone transportation did not affect the quality of TB specimens. The authors recommend additional studies to validate drone transportation of TB specimens over a longer period of time to give further confidence in the adoption of drone delivery in Mozambique.

**Keywords:** TB sputum; quality of sample; ground transportation; drones; Mozambique

## 1. Introduction

Worldwide, there are an estimated 10.4 million new cases of tuberculosis (TB) each year, resulting in 1.7 million deaths [1]. With 552 new cases of TB per 100,000 people per year, Mozambique ranks 14th highest for estimated incidence of TB and 13th highest for incidence of TB/HIV co-infection [2]. Timely identification of new cases is critical to initiating treatment and preventing transmission. It is estimated that one person with active, untreated TB can transmit the disease to as many as 15 other people in one year [3]. As in other low-income countries, missed cases of TB have been attributed in part to difficulties transporting and handling TB specimens; Mozambique is not an exception. The World Health Organization (WHO) recommended GeneXpert molecular testing technology is not available in remote health facilities throughout Mozambique. Instead, sputum microscopy is used in these facilities to screen for potential cases, which are then confirmed through GeneXpert at the district level [3].

There are many challenges that impact the current specimen referral network in Mozambique. Many health facilities are remote or hard-to-reach, requiring a combination of transport methods which can take several hours or even days to reach a district laboratory [4]. Road infrastructure is poor and at times impassable, especially during rainy

season, leaving some health facilities unreachable for many months of the year. These infrastructure challenges can lead to an unreliable and sporadic sample transport system. This results in health workers either referring patients directly to district-level facilities for sample collection or arranging for other means transportation, sometimes out of their own pocket [4].

Given the severity of the TB epidemic, innovative solutions are needed to ensure that patients have access to quality diagnostics services. Medical drone delivery has the potential to address these critical transportation barriers but only if the quality of the samples transported is equal to or better than that of the current transportation method. Drone transportation could adversely affect the quality of specimens since they are subjected to more sudden acceleration, deceleration, vibrations and ambient temperature changes, compared to ground vehicles [5,6].

Countries that have introduced scaled drone delivery into their health systems are starting to conduct and publish research to close a wide evidence gap [7]. Proofs of concept and simulations of drug and vaccine delivery, provision of care technology in emergencies, and transport of laboratory samples and organs via drone have been topics of study [8–14]. In Rwanda, drone transportation has led to faster delivery times for time-critical health products [15].

In Mozambique, drones have been used for aerial photography to support emergency preparedness and disaster response, agricultural productivity, and water conservation, but they have not been tested yet for medical payload delivery [16]. Evidence is needed on the feasibility of drone transportation in Mozambique, including the quality of products transported, effects on supply chain performance and economic viability [17,18]. Failure to do so may create distrust in the technology by laboratory technicians and health workers using the technology [19].

Previous research found that drone transportation had no adverse effect on the time of growth for sputum containing S. aureus and S. pneumoniae pathogens, when transported in a cold climate (3 °C to 8 °C) [5]. Authors recommended that similar studies for other types of organisms, specimens and environmental conditions for further adoption. The purpose of this study is to investigate if the quality of TB specimens transported by drones is equal to or better than ground transportation to understand the feasibility of their use in Mozambique.

## 2. Materials and Methods

One hundred and fifty-six spot sputum specimens were collected in five (5) health facilities from consenting patients. Samples were transported by car to the Instituto Nacional de Saúde (INS) for initial preparation. A mix of positive and negative specimens were then equally divided into one control and one study aliquots. The control samples were transported by ground and the study samples by air using a drone. Flights were performed by an Australian drone logistics provider Swoop Aero. After transportation, the specimens were tested for TB culture (MGIT and Lowenstein–Jensen) in the National TB Reference Laboratory (LNRT).

### 2.1. Study Design

Sample size was calculated based on previous literature on the effect of storage and transport on the cultivability of mycobacteria [20]. Considering a 95% confidence interval, a 90% recovery rate, and a 5% margin of error, the sample size was calculated at 140 positive TB sputum specimens. As it was improbable to obtain 140 positive sputum specimens from suspected patients during the study period (six days), both positive and negative TB specimens were considered. Ultimately, 156 sputum specimens with suspected Mycobacterium tuberculosis were included in the sample.

Study sites for both sample collection and drone flights were identified in collaboration with Mozambique's Civil Aviation Authority (IACM) and Maputo Province Provincial Health Directorate (DPS) using the following criteria:

1.  Adherence to the standards and requirements defined by the Directiva Operacional de Segurança (DOS-09-2018);
2.  An area with semi-rural or rural population density to minimize ground risk;
3.  A geodesic distance of 20–25 km between the laboratory and the health facility;
4.  A close proximity to a reference laboratory for quality analyses;
5.  Approval from DPS.

After obtaining stakeholder agreement and civil aviation approval, the LNRT in Marracuene, Maputo province was selected as the study location. Multiple health facilities for patient sampling were required to obtain the 140 sample size. Based on historical data on the number of samples collected, TB prevalence data and the number of presumptive patients with a TB diagnosis per month, the following health facilities were sampled: Machava II, Matola C, Matola II, Ndlavela and Matola Gare.

For two days prior to study flights, TB specimens were collected at the sample health facilities. Trained health workers received informed consent from study participants. A minimum of 4 mL of sputum was collected from each participant either at the health facility or at home, which is recommended by the national health system. Health technicians supervised on site sample collection. Samples were stored between 2–8 °C in the health facility where the temperature was monitored by digital thermometers and recorded on the respective equipment forms. Samples that did not meet the criteria for the study were discarded as usual by the health unit.

Samples were then sent to LNRT by ground with the study vehicle. Each TB spot sputum specimen was initially processed and divided into two equal aliquots that could be included in the study group and control group, respectively. The following initial analyses were preformed according to LNRT standard operating procedures (POP-DPT-TB-003, rev. 0.7, POP-DPT-TB-004, rev. 0.7 and POP-DPT-TB-056, rev. 0.4):

1.  Macroscopic evaluation and classification on the specimen purulent, mucoid, salivary and hemoptoic qualities;
    -   Review the sample and requisition form to verify that it was the correct analysis requested by the clinician;
    -   Check if the vial containing the sample is labeled with the date and time of collection, if you do not have the time of collection, ask the patient what time it was done;
    -   Remove samples from the transport box and place the box in the disinfection room;
    -   Register samples in the computer system and label the sample with the assigned code, which is maintained throughout the analytical processing to ensure sample and patient traceability;
2.  Mechanical homogenization, to avoid mycobacteria clumping within the specimen, by vortexing with the aid of sterile glass beads;
3.  Division of each sample into 2 aliquots (maximum 4 mL of sputum), place samples in 50 mL falcon tubes and coded each accordingly ("A" for aerial specimens and "T" for ground specimens) with duplicate identification numbers.

After initial processing at LNRT, TB sputum samples were packaged and handled according to the International Air Transport Association (IATA) triple packaging standards and World Health Organization (WHO) guidelines. TB spot sputum samples were placed in rigid waterproof containers with a screw lid. The triple packaged containers were then placed in a Swoop Aero cargo box with ice packs and a temperature sensor, as seen in Figure 1. Study group samples were flown on the pre-approved flight route to Muntanhane and returned to the designated flight site at LNRT, as seen in Figure 2. The cargo box internal temperature was registered through a HOBO MX100 remote temperature monitor and data was received and stored in an Excel database.

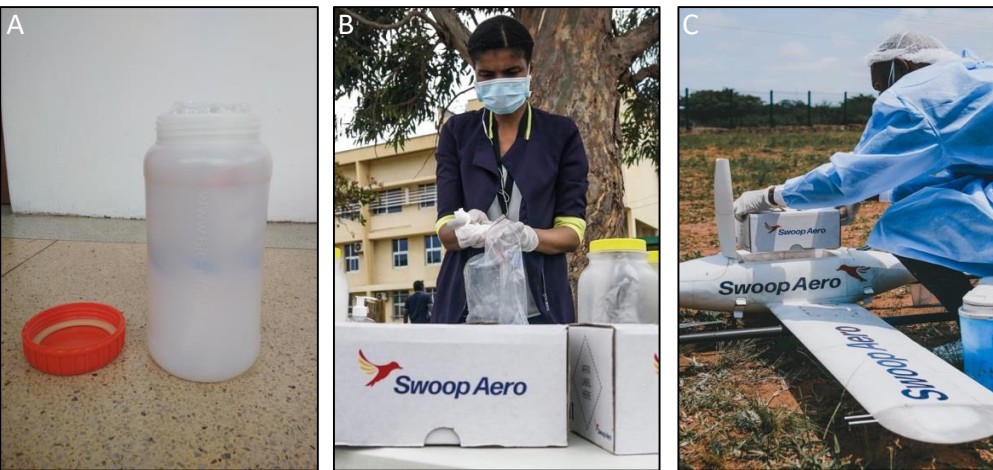

**Figure 1.** Schematic of sample packaging. (**A**) Ridged waterproof container; (**B**) Swoop Aero cargo box; (**C**) Placement of cargo box in aircraft.

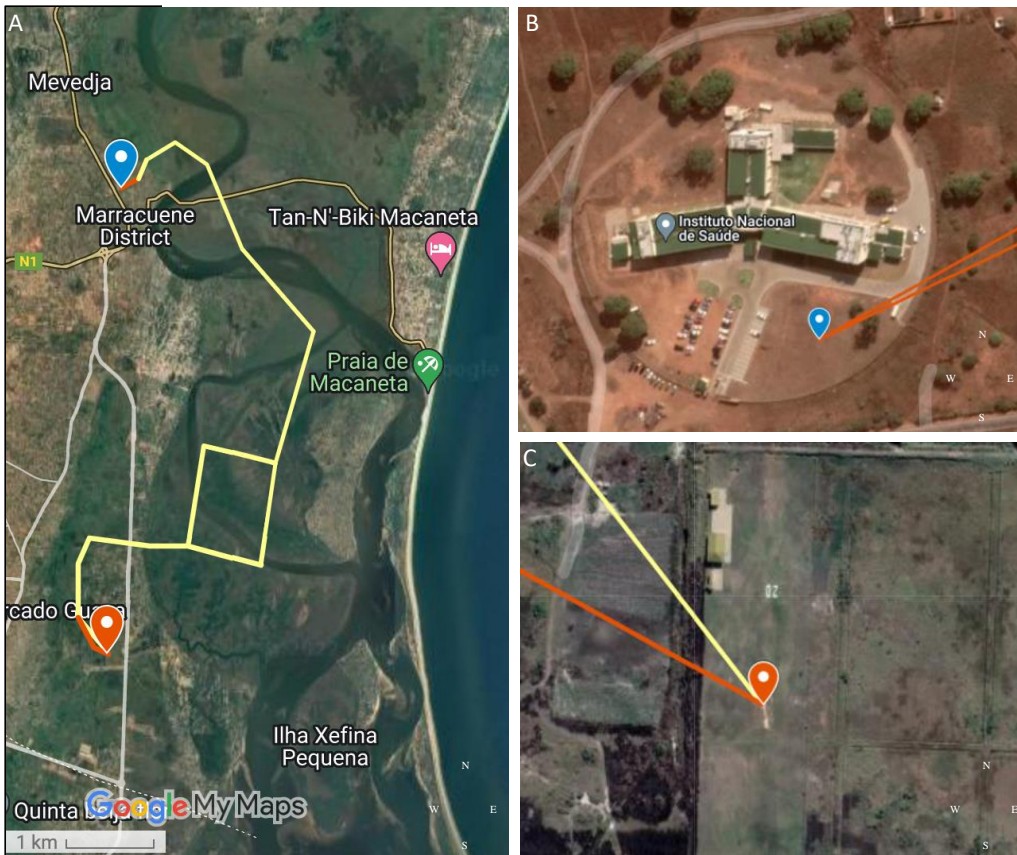

**Figure 2.** Schematic of drone study flight route. (**A**) Overview of flight route; (**B**) Take-off and landing location at LNRT in Marracuene; (**C**) Turn-around point in Muntanhana neighborhood.

The control group samples were packaged following the same procedures and transported by car on a similar route to Muntanhane and returned to the designated flight site at LNRT. The end-to-end workflow of the study is described in Figure 3.

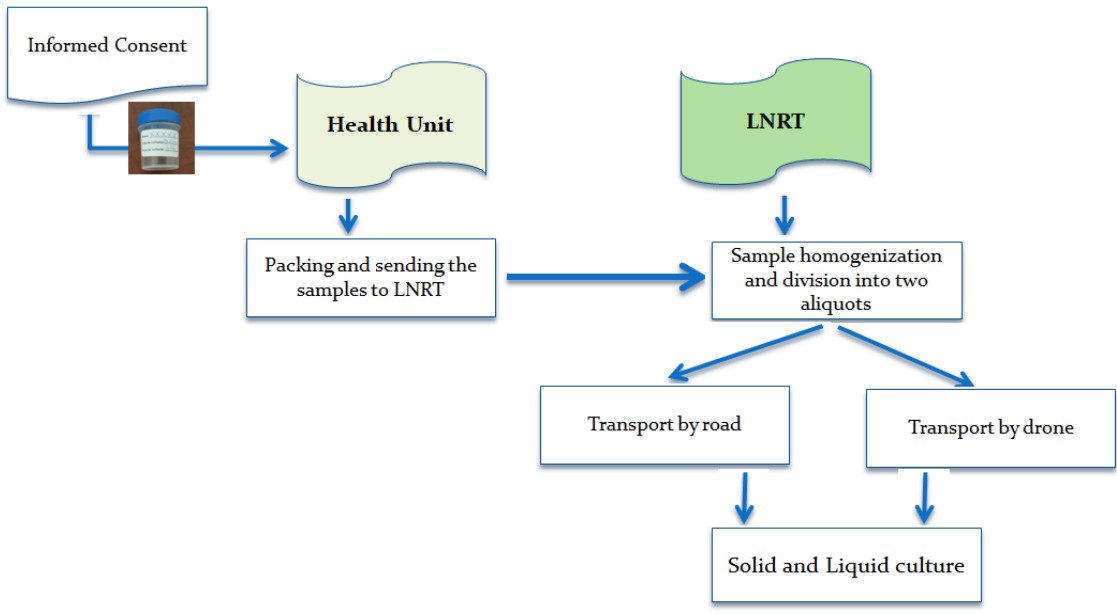

**Figure 3.** Workflow of the study.

*2.2. Sample Analysis*

Both control and study group samples were processed using the NALC-NaOH method. Briefly, a maximum 4 mL of sputum sample was mixed with an equal amount of 5% NALC 6% NaOH, vortexed and incubated at room temperature for 15 min. Samples were then neutralized and washed with a phosphate buffer (pH 6.8) by centrifuging at 3000 g for 15 min at 4 °C. Pellet was resuspended in 2 mL of phosphate buffer and mixed well.

Five hundred microliters of the decontaminated sample was inoculated in Bactec MGIT 960 culture tubes containing a 800 μL mixture of oleic acid, albumin, dextrose, and catalase (OADC) and polymyxin B, amphotericin B, nalidixic acid, trimethoprim, aziocillin (PANTA) supplement as per the manufacturer's instructions (Becton Dickinson Diagnostic Instrument Systems, Sparks, MD, USA), then incubated through equipment which has automated temperature control and 200 μL were inoculated in Lowenstein–Jensen (approximately 3 drops) and incubated at 37 °C in an incubator.

*2.3. Drone Platform*

An Australian drone-powered logistics company, Swoop Aero, was selected as the drone service provider for the study flights through a global Request for Proposal (RFP). Swoop Aero's core technology platform is comprised of the Kookaburra aircraft, detailed in Figure 4 and Table 1, and their proprietary cloud based operations and logistics management software.

**Table 1.** Swoop Aero Platform Specifications.

| Aircraft Attribute | Specification |
|---|---|
| Type | Fixed-wing, hybrid electric with vertical take-off and landing capabilities |
| Max. payload weight | 3 kg |
| Max. payload volume | 5362.5 cm$^3$ |
| Forward speed in flight | 115 km/h |
| Max. distance travelled per charge | 135 km |
| Aircraft battery endurance per charge | 90 min |
| Max. surface temperatures withstood | 55 °C |
| Rain conditions withstood | Safe to operate in light rain (IAW ICAO definition of <2.5 mm/h) indefinitely, and moderate rain (IAW ICAO definition of 2.5–10 mm/h) for up to 30 min |
| Max. wind speed withstood | 50 km/h |

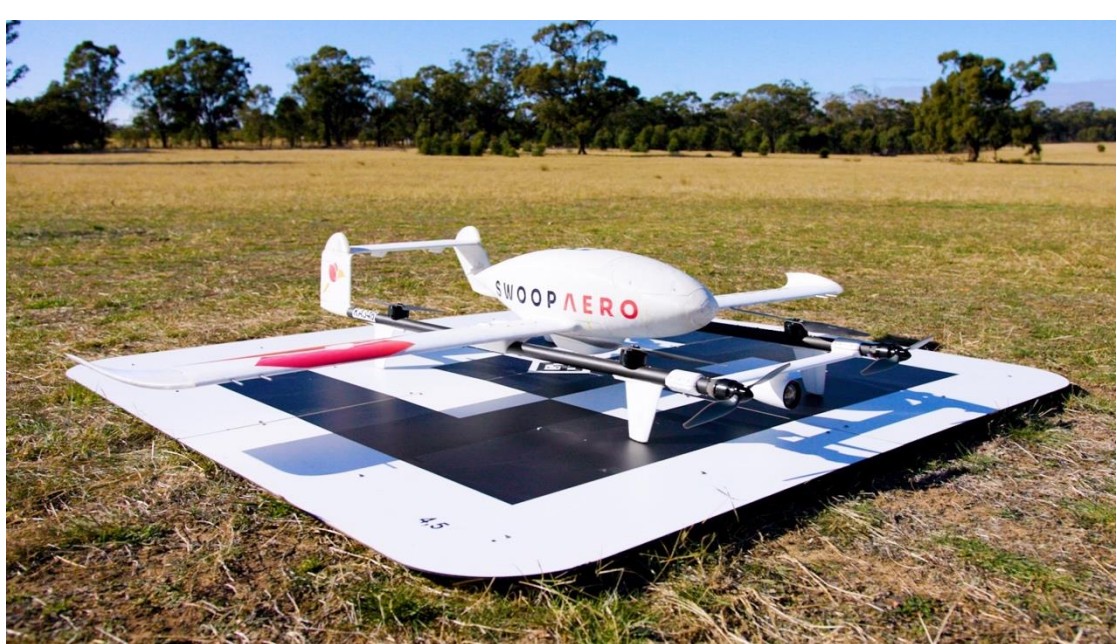

**Figure 4.** Swoop Aero Kookaburra Aircraft.

### 2.4. Ethical and Safety Consideration

The study protocol received ethics approval from the Mozambique National Bioethics Committee and administrative approval from the Mozambique Ministry of Health, with the reference number 492/CNBS/2020. The drone flights were approved by the Civil Aviation Authority with the reference number 0020/DRE/0900/2020 and the Ministry of Defense after an in-depth drone technical and safety application submitted by Swoop Aero and VillageReach with input from an independent aviation expert.

A community awareness campaign was conducted prior to the drone flights to ensure that the general population was aware of the study, its purpose, and instructions for unlikely adverse events.

No personal information about the participants was recorded beyond the informed consent form. The health facility record was not shared with the research team. Samples were identified only with codes and not with personal information. Patient identity was not stored in the study database and all samples were coded to ensure anonymity. All patient were 18 years and above.

Disposal of the biological products was performed as usual with respect to the procedure in place in Mozambique, according to the regulations for safety and disposal of infectious substances.

## 3. Results

### 3.1. Sample Collection

One hundred and fifty-six outpatients with suspected pulmonary TB or who were already under treatment for TB participated in the study. The median age of study participants was 35 years, with 69 (44%) men and 87 (56%) women. There were 125 new patients (80%) and 31 (20%) under treatment. For sample collection, we followed the national guidelines which recommend morning, spot or night specimens. 88 (56%) were early morning, 73 (47%) spot and 52 (33%) night specimens. 107 (69%) of samples were of mucoid type and 49 (31%) were non-mucoid (mucopurulent and saliva). The sample volume ranged from 1.5 mL to 5.0 mL as follows: 27 (1.5 mL to 2.5 mL), 73 (3.0 mL to 3.8 mL) and 56 (4.0 mL to 5.0 mL).

### 3.2. Drone Flights

A total of 18 drone flights were performed over 4 days for a total of 10 flight hours, as seen in Table 2. On average, 8.8 (minimum 6, maximum 10) falcon tubes (50 mL) were transported on each flight for an average of 25 min.

**Table 2.** Study Flight Descriptive Statistics.

| Description | Statistics |
|---|---|
| Total drone flights | 18 |
| Average specimens per flight | 8.8 |
| Average flight speed | 93 km/h |
| Average flight time | 25 min |

Internal cargo box temperature was monitored and registered for all study and control samples. The temperature of the study samples ranged from 3 °C to 8 °C during air transportation while control samples ranged from and 2 °C to 6 °C during ground transportation. No excursions were registered, although the temperature monitoring devices were not validated as a part of this study.

### 3.3. Laboratory Analysis

Specimens were analyzed through the inoculation of liquid (MGIT) and solid culture (LJ) and results were compared respectively in Tables 3 and 4. No significant differences were observed between control and study samples, because we saw 100% agreement for both liquid and solid cultures, as seen in Table 3. Positive TB results were determined by a positive signal given on the BACTEC MGIT 960 machine for liquid cultures. For solid media, positive results were determined by the appearance of colonies on the LJ combined with confirmatory ZN microscopy and Mycobacterium tuberculosis (MCTB) complex (MCTB) specific antigen, MPT64.

**Table 3.** Liquid result for sputum sample.

| Liquid Culture (MGIT) | | Drone | | Total | PPV (%) | NPV (%) |
|---|---|---|---|---|---|---|
| | | Positive | Negative | | | |
| Land | Positive | 19 | 0 | 19 | 100 | 100 |
| | Negative | 0 | 137 | 137 | | |
| | Total | 19 | 137 | 156 | | |
| Agreement | | | 1.0 | | | |

**Table 4.** Solid Media result for sputum sample.

| Solid Culture (LJ) | | Drone | | Total | VPP (%) | VPN (%) |
|---|---|---|---|---|---|---|
| | | Positive | Negative | | | |
| Land | Positive | 4 | 0 | 4 | 100 | 100 |
| | Negative | 0 | 147 | 147 | | |
| | Total | 4 | 147 | 151 | | |
| Agreement | | | 1.0 | | | |

Contamination was detected by fungal growth on a solid culture medium. There was no contamination detected in the liquid culture medium, while for the solid culture medium the contamination rate was 3% for specimens transported by drone (5 samples) and 1% transported by ground (2 samples).

All positive cases identified as MCTB complex presented the same values except for 2 samples with a difference of positivity of 1 day (17%), seen in Table 5. This resulted in no significant difference for time to positivity between the control and study samples.

**Table 5.** Descriptive statistics of the time to positivity for culture of liquid samples transported by drone and by land.

| MGIT Time to Positivity (Days) | Average | Median | Min | Max |
|---|---|---|---|---|
| Drone | 6.2 | 5.55 | 3.08 | 10.09 |
| Land | 5.78 | 5.12 | 3.05 | 10.11 |

## 4. Discussion

Increasingly, drones delivery is seen as a valuable new tool to improve diagnostic sample referral networks but only if it does not adversely affect the samples being transported [21]. Evidence demonstrating vulnerability in the pre-analytical phase has led to a better understanding that secure biological sample transportation is of crucial importance in assuring the reliability of analytical procedures [22,23]. There is very limited real-world data that demonstrate the effect, or lack of effect, of drone transportation on medical specimens [24]. Studies carried out with specimens from real patients with suspected or confirmed disease pathologies are especially limited. To the best of our knowledge, this study is the first published research on the impact of drone transport on the quality of TB specimens. We observed no significant differences between TB sputum samples transported by drone and those transported by road.

100% of results were the same for both samples transported by road and by air. Macroscopic aspects were evaluated to verify if the sample suffered liquefaction during transportation from health facility to LNTR, prior to transport by air or ground. 69% of the samples were good quality sputum specimens (mucoid). This is likely because the vast majority of participants (86%) were patients with new-onset lung pathology and were therefore able to produce a quality sputum sample. Mucus in the respiratory tract is secreted from goblet cells found in the surface epithelium lining the airways and from seromucous glands in the connective tissue layer beneath the mucosal epithelium [25].

After transport by drone or by road, cultures in both liquid and solid medium were conducted to evaluate the higher specificity of solid media and higher sensitivity of liquid media. Positive or negative results and time to positivity were obtained to measure the effects of drone transportation on the bacteria. 12.2% of study samples tested positive for TB. This study found 100% agreement for positive and negative results between control and study samples, for both liquid and solid cultures.

The liquid culture positivity time (MGIT) helps to understand the growth time of Mycobacterium tuberculosis from inoculation to positive detected by the equipment (time to detection [TTD]). To make a semi quantitative assessment of the bacterial load, TTD is analyzed using in the BACTEC MGIT 960 system. The MGIT culture result, along with confirmatory ZN, BAP, and rapid ID tests, are the primary indicators of the presence of viable MCTB in the sputum [26]. In the specimens analyzed, there were no significant differences in TDD between control and study samples. The growth times for the sputum specimens, as well as the colony counts, were similar for the pairs of specimens transported by drone and those transported by ground (6.2 days by drone, 5.78 days by land). There was no significant difference in growth conditions (medium, $O_2$ dependency, etc.) between specimens. Small differences in the amounts of Mycobacterium tuberculosis recovered were registered. Storage and transport may affect the mycobacteria; however, there was growth on all growth media [27].

When cultured in solid medium, study specimens transported by drone showed a contamination rate of 3% while control specimens had a 1% contamination rate. Contamination rates were found to be within the limits established by WHO and were not significant enough between study and control samples to be a significant finding. Literature shows that contamination can be caused by various factors which can influence the culture isolation of MTBC (transport time, sputum appearance, smear grading, location of residence, HIV status, cavitation, age and gender) [28]. As specimens were transported on time, quality specimen collection could be a contributing factor to the minor measured contamination.

### 5. Conclusions

The findings of this study revealed that the agreement between the specimens transported by ground and by drone was acceptable, concluding that drone transportation does not affect the quality of TB specimens. Recruiting enough TB positive participants to reach the calculated sample size was a major challenge. Participants producing the minimally required sputum sample volume (4 mL), one of our inclusion criteria, was challenging. Although 36% of the samples were not 4 mL in volume, this did not affect the quality when they were processed.

The COVID-19 pandemic led to international borders closing in March 2020 just as Swoop Aero was conducting the initial test flights to receive study flight approval from IACM. Swoop Aero was then unable to return to Mozambique until October 2020 for the study flights, requiring special travel approvals from the Australian government. This led to a short duration of study flights (4 days) due to budgetary constraints. Given these promising findings, authors recommend that larger-scale studies are conducted in Mozambique to further validate drone delivery during a longer, real-world implementation. This future research can provide further confidence in the national adoption of drones for sample transportation in Mozambique.

**Author Contributions:** Conceptualization, D.M., L.M. and O.D.; Validation, S.V. and R.B.; Formal analysis, S.V.; Investigation, D.M. and C.M.; Data curation, C.M.; Writing—original draft preparation, D.M.; Writing—review and editing, D.M., S.M., L.M., G.A., O.D. and S.V.; Supervision, S.V.; Funding acquisition, C.L., R.B., L.M. and O.D. All authors have read and agreed to the published version of the manuscript.

**Funding:** This research was funded by United Kingdom Foreign, Commonwealth and Development Office, project number 205074-109.

**Institutional Review Board Statement:** The study was conducted in accordance with the Declaration of Helsinki, and approved by National Ethic Committee of Mozambique and Institutional Ethic for Health in Instituto Nacional de Saúde (Ref: 492/CNBS/20, approved on 9 September 2020).

**Informed Consent Statement:** Informed consent was obtained from all subjects involved in the study.

**Data Availability Statement:** Not applicable.

**Acknowledgments:** The authors wish to express their sincere thanks to the United Kingdom Foreign, Commonwealth and Development Office for their strategy and design support; the Mozambique Civil Aviation Authority on assistance with site selection and ground support during data collection activities and Mozambique Ministry of Health to support in protocol development and data collection.

**Conflicts of Interest:** The authors declare no conflict of interest.

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
