# Peer review of "Quality Analysis of Tuberculosis Specimens Transported by Drones versus Ground Transportation"

_drones, doi:10.3390/drones6070155_

Round 1

Reviewer 1 Report

This paper exploring the quality of TB laboratory specimens after transport via drone, is the newest of a line of papers exploring the stability of biological materials transported by drone. The work and findings are fairly straightforward but the paper needs a good English editor, or a sentence-by-sentence review by its authors. Other issues are smaller and will be detailed later in the review. 

EDITING

In the ABSTRACT, the following sentence, 'A total of 18 drone flights were performed over 4 days (10 tubes of 50mL per flight) and same traveled by ground.' doesn't make clear to the reader whether each tube was 50ml OR whether all 10 tubes combined added up to 50mls. Also 10 tubes per flight * 18 flights = 180 tubes: which is a higher number than the 156 specimens which were collected for the study.

The entire manuscript is littered with these kinds of unclear sentences. The abstract has several. Other major offenders include lines 94, 105, 140 and 237 amongst others. In order for this manuscript to be useful to the readers it will need a thorough editing to clear up these unclear sentences as well as fix syntax issues that are scattered everywhere.

Other Items

  1. Abstract - In addition to the sentence above, the use of sensitivity and specificity in this context and throughout the paper is incorrect. Although we are comparing samples transported by car versus by drone, the TESTs we are using to compare them are the same. I think a more appropriate term is 'Agreement'. 
  2. Introduction - Consider replacing 'the last mile' in line 42 with 'locally'.
  3. Discussion - 
    1. Replace ref. 14 with the paper in the paper in Lancet Global Health describing the implementation of the Rwanda Drone program
    2. From line 242, the line describes the percentage of positive TB samples as a limitation. I think 12% TB positivity would be high in any context and is more than enough to test the hypothesis. I don't think it is a limitation.

Overall this is a good paper that adds to the growing knowledge about the impact of drones on biological specimens. It needs to be cleaned up to be ready for publication.

Reviewer 2 Report

The overall quality of this paper is very good, while there are some English spelling errors. Please go through the manufacture with proofreading. 

Author Response

Response to Reviewer 2 Comments

Point: The overall quality of this paper is very good, while there are some English spelling errors. Please go through the manufacture with proofreading. 

Response: We took note and submitted for english review.

Reviewer 3 Report

This paper describe a study comparing the quality of TB specimens transported by drones and ground transportation. The conclusion was that there was no difference between the two ways of transport. The hypothesis of this study seems to be drone transportation could affect the quality of the TB specimens. A hypothesis testing would, therefore, required. If there was such a hypothesis, the authors should provide relevant scientific basis such as the changes in temperature, vibration, and time of transportation, and so on.  If the author believed there shouldn't be difference between transport by air and on the ground, why bother to do such a study. Without such a study, people would normally assume no difference between transport by air and on the ground.   

Due to the lack of theoretical basis and scientific merits, this paper makes very little contribution, if any, to increase our knowledge in the transportation of biological specimens using a small drone.   

Author Response

Response to Reviewer 3 Comments

Point 1: This paper describe a study comparing the quality of TB specimens transported by drones and ground transportation. The conclusion was that there was no difference between the two ways of transport. The hypothesis of this study seems to be drone transportation could affect the quality of the TB specimens. A hypothesis testing would, therefore, required. If there was such a hypothesis, the authors should provide relevant scientific basis such as the changes in temperature, vibration, and time of transportation, and so on.  If the author believed there shouldn't be difference between transport by air and on the ground, why bother to do such a study. Without such a study, people would normally assume no difference between transport by air and on the ground.

Response: Based on the in-depth TB sample analyses conducted, we believe that lab sample transported by drone was satisfactory and comparable to ground transport. When delivering medicines, samples, the influence of temperature has usually been thoroughly considered. The quality of the sample was not changed during the flight, although temperature and vibration was not validated in this study to see such effect on the samples. The temperatures range were from 3 to 8 ºC for drone and 2 ºC to 6ºC for ground transportation. As we can see, the difference is not much and according to the results obtained, they were similar.

Point 2: Due to the lack of theoretical basis and scientific merits, this paper makes very little contribution, if any, to increase our knowledge in the transportation of biological specimens using a small drone.

Response: One of the hypothesis of this study was to see if drone transportation could affect the quality of the TB specimens as mentioned above. We believe that without such study only theory we could assume normally no difference between transport such sample for TB diagnosis by air and on the ground. But, according to the results of the study we can now see that although temperature, vibration can be a factor who interfere on the quality of medicines, samples, for sputum, the results showed that small drone can be used for biological specimens transportation.

Round 2

Reviewer 3 Report

The authors have made significant revisions both in the background and discussion of the manuscript. There are still some typos needed to be corrected.